# The missing link between genetic association and regulatory function

Noah J Connally[1,2,3]*, Sumaiya Nazeen[1,2,4], Daniel Lee[1,2,3], Huwenbo Shi[3,5], John Stamatoyannopoulos[6], Sung Chun[7], Chris Cotsapas[3,8,9]*, Christopher A Cassa[2,3]*, Shamil R Sunyaev[1,2,3]*

[1]Department of Biomedical Informatics, Harvard Medical School, Boston, United States; [2]Brigham and Women's Hospital, Division of Genetics, Harvard Medical School, Boston, United States; [3]Program in Medical and Population Genetics, Broad Institute of MIT and Harvard, Cambridge, United States; [4]Brigham and Women's Hospital, Department of Neurology, Harvard Medical School, Boston, United States; [5]Department of Epidemiology, Harvard T.H. Chan School of Public Health, Boston, United States; [6]Altius Institute, Seattle, United States; [7]Division of Pulmonary Medicine, Boston Children's Hospital, Boston, United States; [8]Department of Neurology, Yale Medical School, New Haven, United States; [9]Department of Genetics, Yale Medical School, New Haven, United States

*For correspondence:
noahconnally@g.harvard.edu
(NJC);
cotsapas@broadinstitute.org
(CC);
cassa@mit.edu (CAC);
ssunyaev@hms.harvard.edu (SRS)

**Competing interest:** The authors declare that no competing interests exist.

**Abstract** The genetic basis of most traits is highly polygenic and dominated by non-coding alleles. It is widely assumed that such alleles exert small regulatory effects on the expression of *cis*-linked genes. However, despite the availability of gene expression and epigenomic data-sets, few variant-to-gene links have emerged. It is unclear whether these sparse results are due to limitations in available data and methods, or to deficiencies in the underlying assumed model. To better distinguish between these possibilities, we identified 220 gene–trait pairs in which protein-coding variants influence a complex trait or its Mendelian cognate. Despite the presence of expression quantitative trait loci near most GWAS associations, by applying a gene-based approach we found limited evidence that the baseline expression of trait-related genes explains GWAS associations, whether using colocalization methods (8% of genes implicated), transcription-wide association (2% of genes implicated), or a combination of regulatory annotations and distance (4% of genes implicated). These results contradict the hypothesis that most complex trait-associated variants coincide with homeostatic expression QTLs, suggesting that better models are needed. The field must confront this deficit and pursue this 'missing regulation.'

## Editor's evaluation

The findings reported here are important because they address the issue of how complex traits arise from their genetic underpinnings. There is an assumption that genetically mediated variation in transcript abundance, usually detected via analysis of expressed quantitative trait loci, is key to this process, but we lack robust evidence in support of that view. This article finds limited evidence that the baseline expression of trait-related genes explains the associations between complex traits and genetic variants (as identified from genome-wide association studies), leading to the view that the field needs to confront a problem of 'missing regulation.'

## Introduction

Modern complex trait genetics has uncovered surprises at every turn, including the paucity of associations between traits and coding variants of large effect, and the 'mystery of missing heritability,' in which no combination of common and rare variants can explain a large fraction of trait heritability (*Manolio et al., 2009*). Further work has revealed unexpectedly high polygenicity for most human traits and very small effect sizes for individual variants. Enrichment analyses have demonstrated that a large fraction of heritability resides in regions with gene regulatory potential, predominantly tissue-specific accessible chromatin and enhancer elements, suggesting that trait-associated variants influence gene regulation (*Maurano et al., 2012*; *Trynka et al., 2013*; *Gusev et al., 2014*). Furthermore, genes in trait-associated loci are more likely to have genetic variants that affect their expression levels (expression quantitative trait loci, or eQTLs), and the variants with the strongest trait associations are more likely also to be associated with transcript abundance of at least one proximal gene (*Nicolae et al., 2010*). Combined, these observations have led to the inference that most trait-associated variants are eQTLs, and their effects arise from altering transcript abundance, rather than protein sequence. Equivalent sQTL (splice QTL) analyses of exon usage data have revealed a more modest overlap with trait-associated alleles, suggesting that a fraction of trait-associated variants influence splicing, and hence the relative abundance of different transcript isoforms, rather than overall expression levels. The genetic variant causing expression changes may lie outside the locus and involve a knock-on effect on gene regulation, with the variant altering transcript abundances for genes elsewhere in the genome (a *trans*-eQTL), but the consensus view is that *trans*-eQTLs are typically mediated by the variant influencing a gene in the region (a *cis*-eQTL) (*GTEx Consortium, 2020*). Thus, a model has emerged in which most trait-associated variants influence proximal gene regulation.

Here we argue that this unembellished model—in which genome-wide association study (GWAS) peaks are mediated by the effects on the homeostatic expression assayed in tissue samples—is the exception rather than the rule. We highlight the challenges of current strategies linking GWAS variants to genes and call for a reevaluation of the basic model in favor of more complex models possibly involving context-specificity with respect to cell types, developmental stages, cell states, or the constancy of expression effects.

Our argument begins with several observations that challenge the unembellished model. One challenge is the difference between spatial distributions of eQTLs, which are dramatically enriched in close proximity to genes, and GWAS peaks, which are usually farther away (*Stranger et al., 2007*; *Farh et al., 2015*; *Mostafavi et al., 2022*). Another is that expression levels mediate a minority of complex trait heritability (*Yao et al., 2020*). Finally, many studies have designed tools for colocalization analysis: a test of whether GWAS and eQTL associations are due to the same set of variants, not merely distinct variants in linkage disequilibrium. If the model is correct, most trait associations should also be eQTLs, but across studies, only 5–40% of trait associations colocalize with eQTLs (*Giambartolomei et al., 2014*; *Chun et al., 2017*; *Giambartolomei et al., 2018*; *Hormozdiari et al., 2016*).

Despite the doubts raised, the fact that most GWAS peaks do not colocalize with eQTLs cannot disprove the predominant, unembellished model. In a sense, negative colocalization results are confusing because their hypothesis is too broad. If we predict merely that GWAS peaks will colocalize with *some* genes' expression, it is not clear what is meant by a peak's failure to colocalize with *any individual* gene's expression.

Thus, a narrower, more testable hypothesis requires identifying genes we believe a priori are biologically relevant to the GWAS trait. If these trait-linked genes have nearby GWAS peaks and eQTLs, failure to colocalize would be a meaningful negative result. Earlier studies tested all GWAS peaks; when a peak has no colocalization, the model is inconclusive. But trait-linked genes that fail to colocalize reveal that our method for detecting non-coding variation is, with current data, incompatible with our model for understanding it.

With this distinction in mind, we created a set of trait-associated genes capable of supporting or contradicting the model of non-coding GWAS associations acting as eQTLs. For this purpose, the selection of genes becomes extremely important. Because the model attempts to explain the genetic relationship between traits and gene expression, true positives cannot be selected based on measurements of genetic association to traits (GWAS) or expression (eQTL mapping). With this restriction, one source of true positives is to identify genes that are both in loci associated with a complex trait and are also known to harbor coding mutations tied to a related Mendelian trait or the same complex

trait. Using a model not based on expression, Mendelian genes are enriched in common-variant heritability for cognate complex traits (*Weiner et al., 2022*). The genes and their coding variants may be detected in familial studies of cognate Mendelian disorders or by aggregation in a burden test on the same complex phenotypes as GWAS (*Backman et al., 2021*).

For genes whose coding variants can cause detectable phenotypic change, the strong expectation is that a variant of small effect influences the gene identified by its rare coding variants. As an example, *APOE* and *LDLR* are both low-density lipoprotein receptor genes (*Schneider et al., 1981*; *Goldstein and Brown, 1973*). Coding variants in *APOE* and *LDLR* can lead to the Mendelian disorder familial hypercholesterolemia (*Goldstein and Brown, 1973*; *Cenarro et al., 2016*). Even in the absence of a Mendelian coding variant, experiments in animal models have found that the overexpression of these genes reduces cholesterol levels (*Shimano et al., 1992a*; *Shimano et al., 1992b*; *Kawashiri et al., 2001*). GWAS on human subjects have found significant associations near *APOE* and *LDLR*, so it seems reasonable to suspect that any non-coding effects in these loci may be mediated by these genes. This general relationship between Mendelian and complex traits is supported by several lines of evidence summarized in Appendix 1.

## Results

To test the model that trait-associated variants influence baseline gene expression, we assembled a list of putatively causative genes for seven polygenic common traits with available large-scale GWAS data, each of which also has an extreme form in which coding variants of large effect alter one or more genes with well-characterized biology (*Table 1*). Our selection included four common diseases: type II diabetes (T2D) (*Mahajan et al., 2018*), where early-onset familial forms are caused by rare coding mutations (insulin-independent MODY; neonatal diabetes; maternally inherited diabetes and deafness; familial partial lipodystrophy); ulcerative colitis (UC) and Crohn disease (CD) (*Liu et al., 2015*; *Goyette et al., 2015*), which have Mendelian pediatric forms characterized by severity of presentation; and breast cancer (BC) (*Zhang et al., 2020*), where germline coding mutations (e.g., *BRCA1*) or somatic tissue (e.g., *PIK3CA*) are sufficient for disease. We also chose three quantitative traits: low- and high-density lipoprotein levels (LDL and HDL); and height. Between known Mendelian genes and those from *Backman et al., 2021*, our analysis included 220 unique gene-trait pairs (*Figure 1*).

In well-powered GWAS, even relatively rare large-effect coding alleles (mutations in *BRCA1* that cause breast cancer, for instance) may be detectable as an association to common variants, which could make the effect of a coding variant appear to be regulatory instead. To account for this possibility, we computed association statistics in each GWAS locus conditional on coding variants. We applied a direct conditional test to datasets with available individual-level genotype data (height, LDL, HDL); for those studies without available genotype data, we computed conditional associations from summary statistics using COJO (*Yang et al., 2011*; *Yang et al., 2012*; 'Materials and methods'). With both methods, the resulting GWAS associations should reflect only non-coding variants.

After controlling for coding variation, we examined whether these genes are more likely than chance to be in close proximity to variants associated with the polygenic form of each trait. In agreement with existing literature (*Freund et al., 2018*), we observe a significant enrichment for all traits in our combined Mendelian and *Backman et al., 2021* gene sets (*Figure 1—figure supplement 1*).

Of our 220 genes, 147 (67%) fell within 1 Mb of a GWAS locus for the cognate complex trait, over three times as many as the 43 predicted by a random null model (95% confidence interval: 31.5–54.5). Our window of 1 Mb represents roughly the upper bound for distances identified between enhancer–promoter pairs, but most pairs are closer (*Nasser et al., 2021*), so we would expect enrichment to increase as the window around genes decreases; this proves to be the case. At a distance of 100 kb, we find 104 putatively causative genes (47%), though the null model predicts only 11 (95% CI 4.5–17.0), an order-of-magnitude enrichment (*Figure 1—figure supplement 1*). Given their known causal roles in the severe forms of each phenotype, these results suggest that the 147 genes near GWAS signals are likely to be the targets of trait-associated non-coding variants. For example, we see a significant GWAS association between breast cancer risk and variants in the estrogen receptor (*ESR1*) locus even after controlling for coding variation; the baseline expression model would thus predict that non-coding risk alleles alter *ESR1* expression to drive breast cancer risk.

**Table 1.** Putatively causative Mendelian genes.

Each gene includes reference(s) to the known biological role of its coding variants, as established in familial studies, in vitro experiments, and/or animal models. Genes from *Backman et al., 2021* are not included here, but can be found in *Figure 2*.

| Phenotype | Genes |
| --- | --- |
| Low-density lipoprotein | APOB *Soria et al., 1989*; *Pullinger et al., 1995*<br>APOC2 *Hegele et al., 1991*<br>APOE *de Knijff et al., 1994*<br>LDLR *Brown and Goldstein, 1976*<br>LPL *Heizmann et al., 1991*; *Clee et al., 2001*<br>PCSK9 *Abifadel et al., 2003* |
| High-density lipoprotein | ABCA1 *Brooks-Wilson et al., 1999*; *Bodzioch et al., 1999*; *Rust et al., 1999*; *Ordovas et al., 1986*<br>APOA1 *Ordovas et al., 1986*<br>CETP *Glueck et al., 1975*<br>LIPC *Isaacs et al., 2004*; *Grarup et al., 2008*; *Iijima et al., 2008*<br>LIPG *Yamakawa-Kobayashi et al., 2003*<br>PLTP *Jiang et al., 1999*<br><br>SCARB1 *Tai et al., 2003*; *McCarthy et al., 2003* |
| Height | ANTXR1 *Stránecký et al., 2013*; *Bayram et al., 2014*<br>ATR *O'Driscoll et al., 2003*; *Ogi et al., 2012*<br>BLM *Ellis et al., 1995*; *Foucault et al., 1997*<br>CDC6 *Bicknell et al., 2011a*<br>CDT1 *Bicknell et al., 2011a*; *Guernsey et al., 2011*<br>CENPJ *AlDosari et al., 2010*<br>COL1A1 *Wallis et al., 1990*<br>COL1A2 *Spotila et al., 1992*; *De Paepe et al., 1997*<br>COMP *Briggs et al., 1995*; *Mabuchi et al., 2003*<br>CREBBP *Menke et al., 2016*; *Menke et al., 2018*; *Angius et al., 2019*<br>DNA2 *Shaheen et al., 2014*<br>EP300 *Woods et al., 2014*; *Tsai et al., 2011*<br>EVC *Polymeropoulos et al., 1996*; *Ruiz-Perez et al., 2003*<br>EVC2 *Ruiz-Perez et al., 2003*; *Galdzicka et al., 2002*<br>BN1 *Faivre et al., 2003*; *Le Goff et al., 2011*; *Horn and Robinson, 2011*; *Takenouchi et al., 2013*<br>FGFR3 *Hyland et al., 2003*; *Toydemir et al., 2006*; *Makrythanasis et al., 2014*<br>FKBP10 *Alanay et al., 2010*; *Kelley et al., 2011*; *Barnes et al., 2013*<br>HR *Berg et al., 1993*; *Woods et al., 1996*; *Goddard et al., 1995*; *Ayling et al., 1997*<br>KRAS *Aoki et al., 2005*; *Schubert et al., 2006*; *Carta et al., 2006*<br>NBN *Varon et al., 1998*; *Tanzanella et al., 2003*<br>NIPBL *Tonkin et al., 2004*; *Krantz et al., 2004*<br>ORC1 *Bicknell et al., 2011a*; *Guernsey et al., 2011*; *Bicknell et al., 2011b*<br>RC4 *Guernsey et al., 2011*; *Bicknell et al., 2011b*<br>ORC6L *Bicknell et al., 2011a*; *de Munnik et al., 2012*<br>PCNT *Rauch et al., 2008*; *Griffith et al., 2008*; *Piane et al., 2009*<br>PLOD2 *van der Slot et al., 2003*; *HaVinh et al., 2004*; *Puig-Hervás et al., 2012*<br>PTPN11 *Tartaglia et al., 2001*; *Maheshwari et al., 2002*; *Kosaki et al., 2002*<br>RAD21 *Deardorff et al., 2012*; *Kruszka et al., 2019*; *Goel and Parasivam, 2020*<br>RAF1 *Pandit et al., 2007*; *Razzaque et al., 2007*<br>RECQL4 *Lindor et al., 2000*; *Beghini et al., 2003*; *Wang et al., 2003*<br>RIT1 *Aoki et al., 2013*; *Bertola et al., 2014*; *Gos et al., 2014*<br>ROR2 *Afzal et al., 2000*; *van Bokhoven, 2000*; *Tufan et al., 2005*<br>SLC26A2 *Hästbacka et al., 1993*; *Rossi and Superti-Furga, 2001*; *Barreda-Bonis et al., 2018*<br>SMAD4 *Le Goff et al., 2012*; *Caputo et al., 2012*; *Lindor et al., 2012*<br>SRCAP *Hood et al., 2012*; *Le Goff et al., 2013*<br>WRN *Yu et al., 1996*; *Goto et al., 1997*; *Yu, 1997* |

*Table 1 continued*

| Phenotype | Genes |
| --- | --- |
| Crohn disease | ATG16L1 *Hampe et al., 2007* <br> CARD9 *Rivas et al., 2011* <br> IL10 *Fowler, 2005* <br> IL10RA *Gasche et al., 2003*; *Mao et al., 2012* <br> IL10RB *Glocker et al., 2009*; *Begue et al., 2011* <br> IL23R *Duerr et al., 2006*; *Libioulle et al., 2007*; *Glas et al., 2007* <br> IRGM *McCarroll et al., 2008*; *Craddock et al., 2010*; *Prescott et al., 2010* <br> NOD2 *Ogura et al., 2001*; *Hugot et al., 2001* <br> PRDM1 *Ellinghaus et al., 2013* <br> PTPN22 *Diaz-Gallo et al., 2011* |
| Ulcerative colitis | ATG16L1 *Fowler et al., 2008* <br> CARD9 *Rivas et al., 2011* <br> IL23R *Glas et al., 2007*; *Fisher et al., 2008* <br> IRGM *McCarroll et al., 2008* <br> PRDM1 *Ellinghaus et al., 2013* <br> PTPN22 *Diaz-Gallo et al., 2011* <br> RNF186 *Rivas et al., 2016*; *Beaudoin et al., 2013* |
| Type II diabetes | ABCC8 *Reis et al., 2000* <br> BLK *Borowiec et al., 2009* <br> CEL *Bengtsson-Ellmark et al., 2004*; *Raeder et al., 2006* <br> EIF2AK3 *Harding et al., 2001*; *Brickwood et al., 2003*; *Durocher et al., 2006* <br> GATA4 *Shaw-Smith, 2014* <br> GATA6 *Yorifuji et al., 2012*; *De Franco et al., 2013* <br> GCK *Froguel et al., 1993* <br> GLIS3 *Senée et al., 2006* <br> HNF1A *Yamagata et al., 1996b*; *Vaxillaire et al., 1997* <br> HNF1B *Horikawa et al., 1997*; *Lindner et al., 1999* <br> HNF4A *Yamagata et al., 1996a*; *Stoffel and Duncan, 1997* <br> IER3IP1 *Poulton et al., 2011*; *Abdel-Salam et al., 2012*; *Shalev et al., 2014* <br> INS *Støy et al., 2007* <br> KCNJ11 *Hani et al., 1998*; *Gloyn et al., 2004*j <br> KLF11 *Neve et al., 2005* <br> LMNA *Cao and Hegele, 2000* <br> NEUROD1 *Malecki et al., 1999* <br> NEUROG3 *Gradwohl et al., 2000*; *Rubio-Cabezas et al., 2011*; *Pinney et al., 2011* <br> PAX4 *Shimajiri et al., 2001*; *Mauvais-Jarvis et al., 2004*; *Plengvidhya et al., 2007* <br> PDX1 *Stoffers et al., 1997*; *Macfarlane et al., 1999*; *Hani et al., 1999* <br> PPARG *Deeb et al., 1998*; *Savage et al., 2002* <br> PTF1A *Sellick et al., 2004* <br> RFX6 *Smith, 2010*; *Sansbury et al., 2015* <br> SLC19A2 *Labay et al., 1999*, *Oishi et al., 2002*; *Shaw-Smith et al., 2012* <br> SLC2A2 *Laukkanen et al., 2005*; *Sansbury et al., 2012* <br> WFS1 *Strom et al., 1998*; *Hardy et al., 1999*; *Khanim et al., 2001* <br> ZFP57 *Mackay et al., 2008*; *Boonen et al., 2013* |

*Table 1 continued on next page*

*Table 1 continued*

| Phenotype | Genes |
| --- | --- |
| Breast cancer (selected using MutPanning; *Dietlein et al., 2020*) | AKT1<br>ARID1A<br>ATM<br>BRCA1<br>BRCA2<br>CBFB<br>CDH1<br>CDKN1B<br>CHEK2<br>CTCF<br>ERBB2<br>ESR1<br>FGFR2<br>FOXA1<br>GATA3<br>GPS2<br>HS6ST1<br>KMT2C<br>KRAS<br>LRRC37A3<br>MAP2K4<br>MAP3K1<br>NCOR1<br>NF1<br>NUP93<br>PALB2<br>PIK3CA<br>PTEN<br>RB1<br>RUNX1<br>SF3B1<br>STK11<br>TBX3<br>TP53<br>ZFP36L1 |

We next looked for evidence that the trait-associated variants were also altering the expression of our 147 genes in relevant tissues. Controlling for the number of tests we conducted, 134 (91.1%) of these genes had an eQTL in at least one relevant tissue at a false discovery rate (FDR) of Q < 0.05 ('Materials and methods'). If these variants act through changes in gene expression, phenotypic associations should be driven by some of the same variants as eQTLs in relevant tissue types. We therefore looked for co-localization between our GWAS signals and eQTLs in relevant tissues (*Table 2*) drawn from the GTEx Project using three well-documented methods: coloc (*Giambartolomei et al., 2014*), JLIM (*Chun et al., 2017*), and eCAVIAR (*Hormozdiari et al., 2016*). We found support for the colocalization of trait and eQTL association for only 7 genes out of 147 (4.8%) for coloc; 10/147 (6.8%) for JLIM; and 8/147 (5.4%) for eCAVIAR. Accounting for overlap, this represents only 18/220 putatively causative genes (8.2%) or 18/147 (12.2%) putatively causative genes near GWAS peaks, even without full multiple-hypothesis testing correction ('Materials and methods'), which is not obviously better than random chance. We note that prior estimates of the fraction of *GWAS associations* colocalizing with eQTLs (25–40%; *Giambartolomei et al., 2014*; *Chun et al., 2017*; *Hormozdiari et al., 2016*; *Wen et al., 2017*) do not directly evaluate the ability to find causative genes. By contrast, our estimate of the number of putatively causative *genes* that colocalize with eQTLs tests the consistency of our knowledge, models, and data.

A potential weakness of our approach is the restriction of our search to predefined tissues. We believe this is necessary in order to avoid the disadvantages of testing each gene–trait pair in each tissue—either a large number of false positives or a severe multiple-testing correction that may lead to false negatives. However, restricting to the set of tissues with a known biological role and available expression data almost certainly leaves out tissues with relevance in certain contexts. Some of the tissues we do use have smaller sample sizes, limiting their power to detect eQTLs with smaller effects.

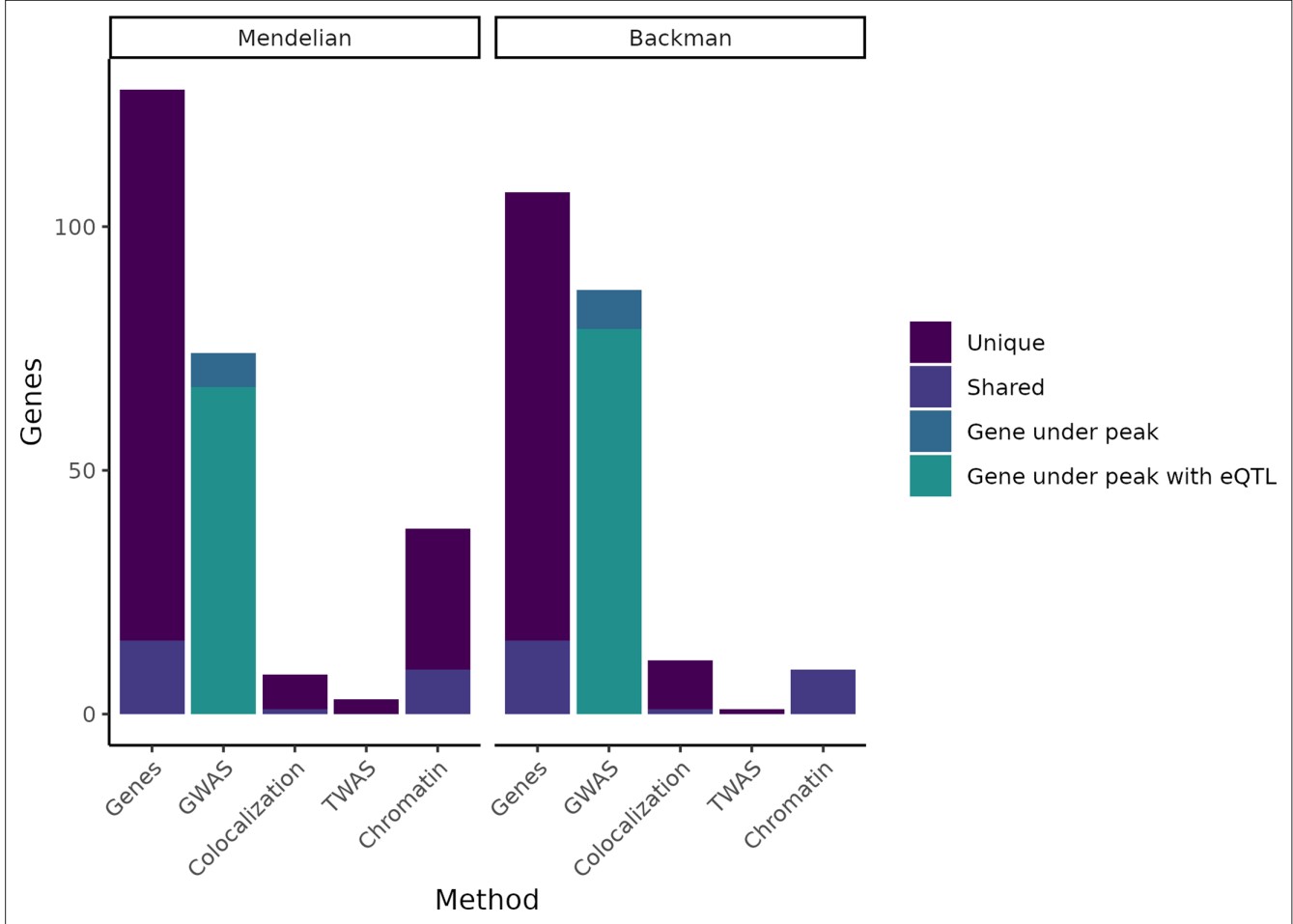

**Figure 1.** Putatively causative genes identified by each method category. The leftmost column in each half of the plot displays the entire group of putatively causative genes for our Mendelian set of genes and our (*Backman et al., 2021*) set of genes, respectively, as well as noting how many are unique to each set or shared between the two sets. The second column in each half indicates how many genes from each set have a nearby GWAS peak or have both a nearby GWAS peak and an expression QTL (eQTL). The remaining columns indicate how many genes were identified through colocalization, transcriptome-wide association studies (TWAS), or chromatin methods, while noting how many of these genes are unique vs. shared between the Mendelian and Backman sets.

The online version of this article includes the following figure supplement(s) for figure 1:

**Figure supplement 1.** Enrichment of Mendelian genes near GWAS peaks.

**Figure supplement 2.** Change in coloc hits when adjusting expression QTL(eQTL) statistics using Multivariate Adaptive Shrinkage Method (MASH).

To address potential shortcomings from the available sample of tissue contexts, we incorporated the Multivariate Adaptive Shrinkage Method (MASH) (*Urbut et al., 2019*). MASH is a Bayesian method that takes genetic association summary statistics measured across a variety of conditions and, by determining patterns of similarity across conditions, updates the summary statistics of each individual condition. In our case, if an eQTL is difficult to find in a tissue of interest, incorporating information from other tissues may help us detect it. Unlike meta-analysis, this method generates summary statistics that still correspond to a specific tissue.

We ran MASH on every locus used in our earlier analysis using data from all non-brain GTEx tissues ('Materials and methods'). Rerunning coloc with these modified statistics increased the number of GWAS-eQTL colocalizations across all genes by 26% (from 389 to 489). However, the 100 new colocalizations identified only four additional putatively causative genes (*Figure 1—figure supplement 2*). These results indicate that tissue-type selection was not the limiting factor in our analysis.

Transcriptome-wide association studies (TWAS) (*Gamazon et al., 2015*; *Gusev et al., 2016*; *Mancuso et al., 2017*; *Barbeira et al., 2018*) are another class of methods applied to identify causative

**Table 2.** Tissue-trait pairs.
Tissues were selected for each trait based on a priori knowledge of disease biology.

| Mendelian trait | GWAS trait | Tissues examined |
|---|---|---|
| Breast cancer | Breast cancer | Breast mammary tissue |
| Crohn disease | Crohn disease | Small intestine terminal ileum Colon sigmoid Colon transverse |
| Ulcerative colitis | Ulcerative colitis | Small intestine terminal ileum Colon sigmoid Colon transverse |
| Dyslipidemia Hyperlipidemia Tangier's disease | High-density lipoprotein | Liver Adipose (subcutaneous) Whole blood |
| Dyslipidemia Hyperlipidemia | Low-density lipoprotein | Liver Adipose (subcutaneous) Whole blood |
| Mendelian short stature | Height | Skeletal muscle |
| Monogenic diabetes | Type II diabetes | Pancreas Skeletal muscle Adipose (subcutaneous) Small intestine terminal ileum |

genes under GWAS peaks using gene expression. TWAS measures genetic correlation between traits and is not designed to avoid correlations caused by LD, which gives it higher power in the case of allelic heterogeneity or poorly typed causative variants (*Wainberg et al., 2019*). However, while sensitive, TWAS analyses typically yield expansive result sets that include many false positives and are sensitive to the number of tissue types (*Wainberg et al., 2019*). Results from the FUSION implementation of TWAS (*Mancuso et al., 2017*) across all tissues identified our putatively causative genes as likely tied to the GWAS peak in 66/220 loci (30%). However, only 4/220 (1.8%) genes were identified by FUSION when we restricted the analysis to relevant tissues.

Given the paucity of expression-mediated GWAS peaks, we asked whether GWAS variants in our study loci reside in likely regulatory sites. Taking the 128 genes in the Mendelian subset of putatively causative genes, we fine-mapped each nearby GWAS association using the SuSiE algorithm (*Wang et al., 2020b*). For 37 of these genes, we identified at least one high-confidence fine-mapped variant (defined as a variant with posterior inclusion probability, or PIP, greater than 0.7) within 100 kb of the transcription start site. We tested whether these fine-mapped variants fall within (i) regulatory DNA marked by DNase I hypersensitive sites (DHS) *Meuleman et al., 2020*; (ii) a narrowly mapped active histone modification feature (H3K27ac, H3K4me1, or H3K4me3; *Nasser et al., 2021*); or (iii) sites marked as an 'enhancer' by ChromHMM (*Ernst and Kellis, 2012*; *Ernst and Kellis, 2017*; 'Materials and methods'). As many as 32/37 (86%) genes identified this way have a fine-mapped variants within a candidate regulatory region across all the tissue types examined, or 25/37 (68%) when restricting to trait-relevant tissues (*Figure 2*); (*Supplementary file 1* and *Supplementary file 2*). Despite this strong evidence that these GWAS associations arise from effects on regulatory DNA, only 5/25 loci (20%) demonstrably correspond to expression effects in our eQTL analysis.

In order to more analogously compare our regulatory feature analysis to our eQTL analysis, we computed 'activity-by-distance' (ABD)—a simplification of the 'activity-by-contact' method that provides a measure connecting a distal regulatory region to a target gene (*Nasser et al., 2021*; *Fulco et al., 2019*; 'Materials and methods' *Figure 3*). Taking each locus's feature with the highest ABD score, we implicate 5/37 (14%) of our Mendelian subset of genes. As such, even when a GWAS association and trait-relevant gene are in the same locus, they are difficult to link, whether using eQTLs or current approaches to integrating chromatin data with target genes.

## Discussion

Overall, our results are strongly consistent with the idea that complex traits are governed by noncoding genetic variants whose effects on phenotype are mediated by their contribution to the regulation of nearby genes. However, these same results are inconsistent with a model that such effects on gene regulation arise from the genetic influences on baseline gene expression that are captured by eQTLs.

The enrichment of putatively causative genes—selected based on existing biological knowledge—near GWAS peaks supports their role in complex traits. Additionally, the enrichment of fine-mapped GWAS variants in likely regulatory regions marked by DHS and other chromatin features lends support

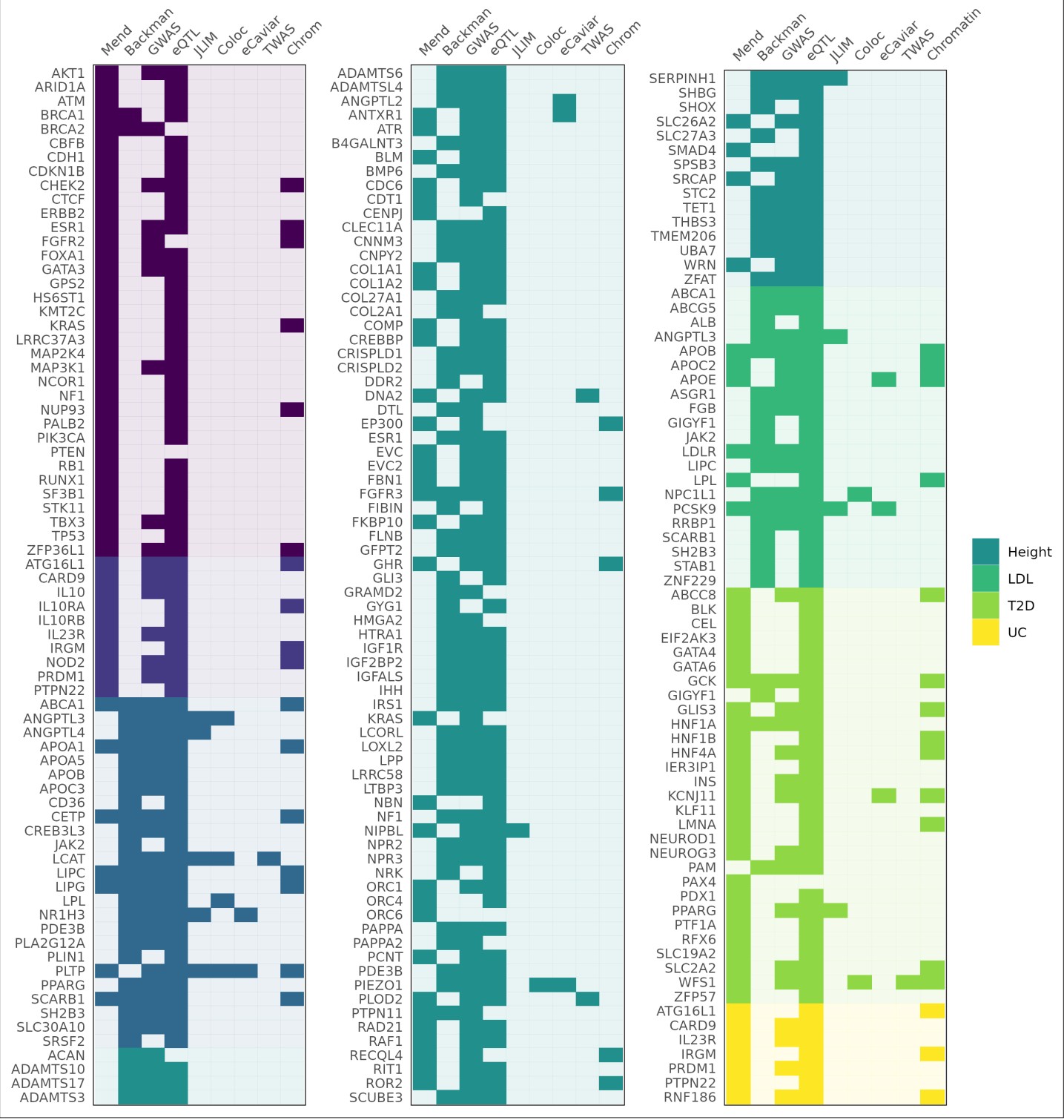

**Figure 2.** Genes identified as associated with a complex trait by each method. Columns 'Mend' and 'Backman' indicate whether a gene is from the Mendelian set of putatively causative genes, the Backman et al. set, or both. Subsequent columns indicate whether a gene was identified as a hit using each of our methods: JLIM, coloc, eCaviar, transcriptome-wide association studies (TWAS), and chromatin analysis.

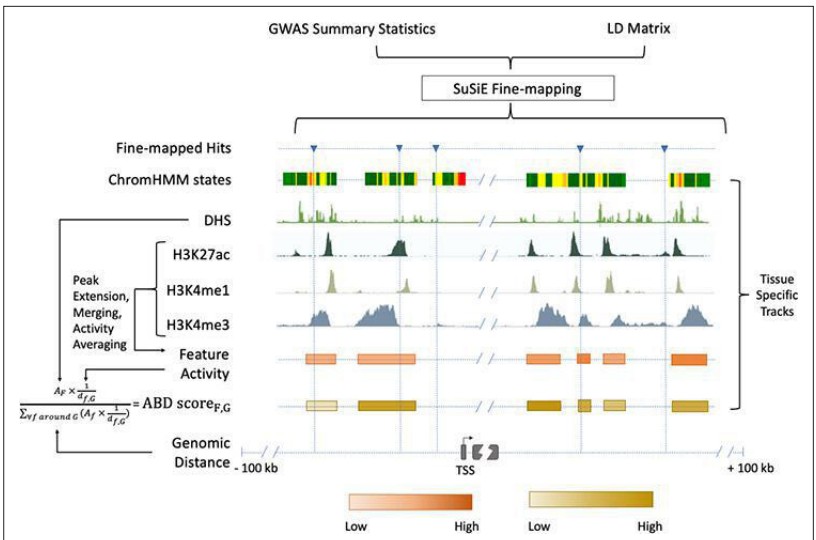

**Figure 3.** Chromatin-based causative gene identification. Following the fine-mapping of GWAS variants, three parallel methods were used. The first identified fine-mapped variants falling within regions annotated as enhancers by ChromHMM. The second identified variants within histone modification features and evaluated their relevance using an activity-by-distance (ABD) score that combined the strength of the feature (i.e., the strength of the acetylation or methylation peak) with its genomic distance to the gene of interest ('Materials and methods'). The third repeated both of these—checking for fine-mapped variants within a region and calculating the ABD score—for DNase I hypersensitivity sites.

The online version of this article includes the following source data for figure 3:

**Source data 1.** Gene-level results for linked expression and traits.

to the general model of GWAS associations arising from effects on gene regulation and expression. However, the inability of varied statistical methods to connect GWAS associations and expression argues against the idea that the causative GWAS variants exert their effects via the homeostatic or blended effects on gene expression that are captured by bulk-tissue eQTLs of the sort discovered by broad expression-data collection projects.

Many explanations have been suggested for the limited success of expression-centric methods to explain the mechanisms of GWAS variants. Undirected, broad approaches—including most GWAS-eQTL linking studies—are designed to be largely independent of a priori biological knowledge and hypotheses. This unconstrained focus is ideal for discovery because it delivers the largest number of positive findings, but it is ill-suited for providing explanations for negative results—when you do not know what you were looking for, it is hard to explain why you did not find it. By testing only loci for which there is a strongly suspected contributing gene, we were better able to distinguish which factors might prevent us from identifying GWAS-to-gene links using expression alone.

As a result, we conclude that a number of explanations often considered when evaluating expression-based variant-to-gene methods are not applicable in the context we examined. These include (i) non-expression-mediated mechanisms, (ii) lack of statistical power for GWAS, (iii) the absence of eQTLs for relevant genes, (iv) and underpowered methods for linking expression to GWAS (*Table 3*).

Our results suggest that there remains to be discovered a major component connecting variants to gene expression, which we term 'missing regulation.' We propose that this currently lacking component could be exposed by considering more nuanced experimental and analytical models of gene expression. One likely model involves context-dependent gene expression—expression whose relevant effects are confined to (i) particular refined cell types or anatomical partitions, or (ii) cell states such as responses to perturbations, whether exogenous (e.g., environmental change) or endogenous (e.g., cell differentiation). A complementary model may incorporate heterogeneity of gene expression or the variance of expression across relatively short time scales. These models and others may depend on or be augmented by thresholding or buffering of variant effects, which may produce changes in

**Table 3.** Proposed explanations for negative results under the unembellished model.
Many explanations have been proposed for GWAS associations that are not explained by *cis*-QTLs. This table details the explanations inconsistent with our results, which are explained in the left column and addressed on the right. Explanations involving more detailed models of gene regulation can be found in *Table 4*. Two of the explanations addressed here involve violations of the assumptions of our and other expression-based complex trait studies. If coding and non-coding variants affect fundamentally different biological pathways, or if trait associations rarely depend on *cis*-eQTLs, our methods of mapping regulation to traits would have nothing to uncover. Even in the presence of eQTL-driven trait associations, insufficient power to detect trait associations, to detect eQTL associations, or to link the two would result in predominantly negative results.

| Violated assumptions | |
| --- | --- |
| Genes implicated via coding variants are irrelevant for non-coding associations | • Our genes are enriched for GWAS associations even after removing the effects of coding variants<br>• Loss-of-function variants, which underlie many Mendelian-trait genes, can be thought of as large-effect eQTLs<br>• Genes identified from *Backman et al., 2021* are not based on cognate phenotypes, but the same complex phenotypes as GWAS |
| Regulatory mechanisms other than *cis*-eQTLs | • Splice QTLs are consistently found to explain less phenotypic variance than eQTLs, and they cannot explain the many GWAS associations that fall within intergenic regions<br>• *Trans*-eQTLs are believed to rely on their effects as *cis*-eQTLs for other genes; the few exceptions to this model (e.g., CTCF binding sites) are not broadly applicable |
| **Insufficient power** | |
| Lack of GWAS power | • GWAS have been shown to have sufficient power to identify small effects even in rare variants<br>• 2/3 of the genes we used have nearby GWAS associations, reflecting a strong enrichment and indicating that GWAS discovery is not a limiting factor<br>• Our analysis is conditioned on the presence of GWAS associations |
| Lack of eQTL mapping power | • GTEx is well powered for eQTL discovery in bulk tissue *GTEx Consortium, 2020*; *Gamazon et al., 2015*<br>• 93% of our genes have a mapped cis-eQTL in a relevant tissue |
| Lack of power for colocalization and TWAS methods | • Simulations show that colocalization and TWAS methods are well-powered *Chun et al., 2017*; *Giambartolomei et al., 2018*; *Hormozdiari et al., 2016*; *Gusev et al., 2016*; *Hukku et al., 2021*<br>• They are robust to levels of LD mismatch higher than what would be expected given our datasets *Chun et al., 2017*; *Wainberg et al., 2019*; *Hukku et al., 2021*<br>• Some, though not all, of the methods are robust to allelic heterogeneity *Chun et al., 2017*; *Wainberg et al., 2019* |

eQTL = expression QTL; TWAS = transcriptome-wide association studies.

gene expression with nonlinear effects on phenotype. A summary of proposed models can be found in *Table 4*.

We propose that finding the 'missing regulation' will require not only identifying novel eQTLs explaining GWAS peaks, but also explaining the phenotypic irrelevance of 'red herring eQTLs'—that

**Table 4.** Explaining negative results with more nuanced models of gene regulation.
To reconcile an expression-based model with our observations requires us to both explain the absence of trait-linked eQTLs as well as explaining away the inconsequence of eQTLs for trait-linked genes. The left-hand side lists additions or changes to the unembellished model, while the right-hand side contains explanations of the models and current relevant research.

| Extended models of gene regulation | |
| --- | --- |
| Context dependency: a context-specific eQTL, invisible in bulk tissues analyzed to date, replaces or supplements the bulk tissue homeostatic eQTL | Cell type *Dobbyn et al., 2018*; *Zhang et al., 2018*; *Schmiedel et al., 2018*; *Glastonbury et al., 2019*; *Rai et al., 2020*; *Findley et al., 2021*; *Neavin et al., 2021*; *Ota et al., 2021*; *Patel et al., 2021*; *Bryois et al., 2021*; *Arvanitis et al., 2022*; *Oelen et al., 2022*; *Perez et al., 2022*; *Schmiedel et al., 2022*; *Yazar et al., 2022* |
| | • Only a subset of cell types in the tissue contribute to the GWAS phenotype. <br> • An eQTL specific to such a cell type is causative for the phenotype. <br> • The eQTL either cannot be detected in bulk tissue because of the cell type's low prevalence <br> • The appropriate cellular or anatomical context has not yet been analyzed |
| | Developmental timing *Dobbyn et al., 2018*; *Strober et al., 2019*; *Cuomo et al., 2020*; *Bonder et al., 2021*; *Jerber et al., 2021*; *Aygün et al., 2022*; *Elorbany et al., 2022* |
| | • The GWAS phenotype depends on a specific point in cell/tissue development or differentiation <br> • eQTLs present at the correct interval contribute to phenotype, but eQTLs observed at other points do not |
| | Cell state or environment *Findley et al., 2021*; *Ota et al., 2021*; *Oelen et al., 2022*; *Schmiedel et al., 2022*; *Huh and Paulsson, 2011*; *Knowles et al., 2017*; *Kim-Hellmuth et al., 2017*; *Balliu et al., 2021*; *Mu et al., 2021*; *Ward et al., 2021*; *Nathan et al., 2022*; *Baca et al., 2022* |
| | • The causative eQTL has effects that are undetectable in steady-state expression under normal conditions <br> • It may activate only in response to a specific environmental condition, such as immune activation or a metabolic shift |

*Table 4 continued on next page*

*Table 4 continued*

| | Extended models of gene regulation |
|---|---|
| | Nonlinearity *Fu et al., 2009*; *Dori-Bachash et al., 2011*; *Ghazalpour et al., 2011*; *Pai et al., 2012*; *Vogel and Marcotte, 2012*; *Khan et al., 2013*; *Wu et al., 2013*; *McManus et al., 2014*; *Albert and Kruglyak, 2015*; *Bader et al., 2015*; *Battle et al., 2015*; *Cenik et al., 2015*; *McManus et al., 2015*; *Pai et al., 2015*; *Schafer et al., 2015*; *Chick et al., 2016*; *Liu et al., 2016*; *Schaefke et al., 2018*; *Buccitelli and Selbach, 2020*; *Wang et al., 2020a*; *Kusnadi et al., 2022* |
| | • There may be buffering that prevents a change in expression from producing a change in protein levels<br>• Expression below a certain level may not influence phenotype, rendering small eQTLs irrelevant |
| | Steady-state expression may be a poor model *Pedraza and Paulsson, 2008*; *Raj and van Oudenaarden, 2008*; *Shahrezaei and Swain, 2008*; *Larson et al., 2009*; *Raj and van Oudenaarden, 2009*; *Suter et al., 2011*; *Dar et al., 2012*; *Viñuelas et al., 2013*; *Kumar et al., 2015*; *Nicolas et al., 2017*; *Qiu et al., 2019*; *Wang et al., 2020c* |
| Nonlinear or non-homeostatic: the relationship between eQTL and genotype is indirect | • Phenotype may depend on the kinetics of expression, which could be cyclical or follow some other pattern<br>• Expression may be stochastic, such that only a random subset of cells display the relevant expression pattern at any one time |

eQTL = expression QTL.

is, eQTLs for putatively causative genes that fall near GWAS peaks but do not colocalize with them. Above, we use the example genes *APOE* and *LDLR*. Both these genes harbor coding variants causing Mendelian hypercholesterolemia, and both have non-coding variants that GWAS have tied to LDL levels. Both have eQTLs in trait-relevant tissues. For *APOE*, these points cohere into an explanation: the LDL association is an eQTL for the lipid-binding gene. But for *LDLR*—and for most genes—the association, the mechanism, and the gene cannot be tied together.

Importantly, our results do not diminish the importance or general utility of eQTLs. Rather, they suggest that current models are deficient in two respects: (i) they fail to unify trait-associated non-coding variants with known trait-associated genes, and (ii) they fail to explain the non-effects of identified 'red herring' eQTLs. These deficiencies highlight a need for new approaches to the role of gene regulation in complex traits.

One long-standing goal of GWAS has been to discover genes contributing to complex traits (*Manolio et al., 2009*; *Farh et al., 2015*), but low rates of positive findings for expression-based variant-to-gene methods have constrained this possibility (*Chun et al., 2017*; *Baird et al., 2021*). Among other challenges, this has limited the benefit of GWAS and expression data for disease-gene mapping and drug discovery (*Baird et al., 2021*; *Umans et al., 2021*). Another practical question raised is the value of current large-scale public datasets. Compared to genotypes, expression data are relatively difficult to collect, particularly from specialized cell contexts. If the most relevant models are shown to depend on effects not observable in homeostatic eQTL mapping, the field may need to consider prioritizing other biological contexts and forms of expression data.

# Materials and methods
## Gene selection
By manual literature search, we selected 128 genes harboring large-effect-size coding variants for one of the seven phenotypes (*Table 1*; specifically, we selected 128 gene–trait pairs, representing 121

unique genes). These genes were identified using familial studies, rare disease exome-sequencing analyses, and, for breast cancer, using the MutPanning method (*Dietlein et al., 2020*) (citations for each gene are included in *Table 1*). Review papers, as well as the OMIM database (*McKusick-nathans institute of genetic medicine, 2021*), were generally used as starting points, but an examination of the original literature was needed to confirm genes' suitability. For example, though SMC3 is known to cause Cornelia de Lange syndrome, which is characterized in part by short stature, SMC3 mutations lead to a milder form of the syndrome, usually without a marked reduction in stature (*Deardorff et al., 2007*). Several of these phenotypes—height, HDL, cholesterol, breast cancer, and type II diabetes—were also analyzed in *Backman et al., 2021*, which, through burden testing, identified a total of 110 genes; after accounting for overlaps, this increased our set of putatively causative genes to 220.

The inclusion of genes from Backman et al. ensures that our results are not dependent on an undetected bias in our selection. The set of genes chosen from familial studies offered the advantage that it was selected based on independent methods and data distinct from the large-scale genotyping studies that have characterized the GWAS era. The tradeoff to this was the impossibility of selecting genes through a fully systematic and non-arbitrary process. Because this work was performed in the UKBB, there is some overlap between their data and ours. However, our work did not use exomes, and most of the variants driving their findings are too rare to influence GWAS results. When this is not the case, our decision to condition on coding variants should make the effects used in our work independent from their findings.

## Identifying coding variants

Because GWAS sample sizes are large enough to detect the low-frequency coding variants used to select some of our genes, it is possible that a coding variant would distort the association signal of nearby eQTLs. To minimize this concern, we removed the effects of coding variants on GWAS. Many variants can fall within coding sequences in rare splice variants, so it is important to remove only those variants that appear commonly as coding. These coding variants were selected based on the pext (proportion of expression across transcripts) data (*Cummings et al., 2020*). Two filters were used. First, we removed genes whose expression in a trait-relevant tissue was below 50% of their maximum expression across tissues. Second, we removed variants that fell within the coding sequence of less than 25% of splice isoforms in that tissue. The remaining variants were used to correct GWAS signal, as explained below. The code for this analysis, and all other quantitative analyses in this paper, can be accessed at https://github.com/NJC12/missing_link_association_function (*NJC12, 2023*; copy archived at swh:1:rev:46d9072b7cc13f6532203d1494eec4d0f634e092).

## GWAS

For height, LDL cholesterol, and HDL cholesterol, GWAS were performed using genotypic and phenotypic data from the UKBB. In order to avoid confounding, we restricted our sample to the 337K unrelated individuals with genetically determined British ancestry identified by *Bycroft et al., 2017*. The GWAS were run using Plink 2.0 (*Chang et al., 2015*), with the covariates age, sex, body mass index (for LDL and HDL only), 10 principal components, and coding variants.

## Conditional analysis

Because UKBB has limited power for breast cancer, Crohn disease, ulcerative colitis, and type II diabetes, we used publicly available summary statistics. The Conditional and Joint Analysis (COJO) (*Yang et al., 2011*; *Yang et al., 2012*) program can condition summary statistics on selected variants—in our case, coding variants—by using an LD reference panel. For this reference, we used TOPMed subjects of European ancestry (*Taliun et al., 2021*). The ancestry of these subjects was confirmed with FastPCA (*Galinsky et al., 2016a*; *Galinsky et al., 2016b*), and the relevant data were extracted using bcftools (*Danecek et al., 2021*). Our conditional GWAS data are available at doi:10.5061/dryad.612jm644q.

## Enrichment analysis

At each distance, the number of Mendelian and non-Mendelian genes within that window around GWAS peaks are counted. p-Values were calculated using Fisher's exact test (*Figure 1*, *Figure 1—figure supplement 1*). Because Mendelian genes may be unusually important beyond our chosen traits, we conduct a set of controls by measuring the enrichment of non-matching Mendelian and

complex traits (CD genes and BC GWAS; BC genes and LDL GWAS; LDL genes and UC GWAS; UC genes and height GWAS; height genes and T2D GWAS; T2D genes and HDL GWAS; HDL genes and CD GWAS).

## eQTL detection

eQTL summary statistics were taken from GTEx v7. Some methods detect colocalization with variants that are individually significant, but would not pass a genome-wide threshold (*Chun et al., 2017*). Because we tested only a subset of genes, we used the Benjamini–Hochberg method (*Benjamini and Hochberg, 1995*) to calculate the FDR based on the number of tests we conducted multiplied by a correction factor to account for variants that are tested in combination with a gene but are not reported (a factor of 20 closely matched the genome-wide FDR results for GTEx). With this method, 204/220 (93%) of our genes displayed an eQTL, including 134/147 genes with a nearby GWAS peak (91%). Even using the FDR statistics of the GTEx project—which are based on the assumption of testing every gene in every tissue—107/220 (49%) of our genes and 76/147 (52%) of genes near GWAS peaks had an eQTL at Q < 0.05.

## Colocalization

JLIM (*Chun et al., 2017*) was run using GWAS summary statistics and GTEx v7 genotypes and phenotypes for the tissues listed in *Table 2*. Coloc (*Giambartolomei et al., 2014*) was run using GWAS and GTEx v7 summary statistics for the same tissues. eCAVIAR (*Hormozdiari et al., 2016*) was run using GWAS and GTEx v7 summary statistics for these tissues, and a reference dataset of LD from UKBB (*Weissbrod et al., 2020*). MASH was run incorporating data from all non-brain tissues, and coloc was rerun using the adjusted values for the same tissues as before.

## MASH

MASH was applied to all GTEx tissues using the mashr R-package (*Urbut et al., 2019*). We restricted this model to non-brain tissues—which include all of our trait-selected tissues—due to the known tendency of brain and non-brain tissues to cluster separately in expression analysis (*Battle et al., 2017*; *Park et al., 2017*; *Gamazon et al., 2019*).

## Fusion (TWAS)

We used the FUSION implementation of TWAS, which accounts for the possibility of multiple *cis*-eQTLs linked to the trait-associated variant by jointly calling sets of genes predicted to include the causative gene, to interrogate our 220 loci (*Mancuso et al., 2017*). FUSION included our putatively causative genes in the set identified as likely relevant to the GWAS peak in 66/220 loci (30%). However, interpretation of this TWAS result is difficult. For many complex traits, TWAS returns a large number of findings (e.g., >150 for LDL cholesterol and >4800 for height). This is in part due to the multiple genes jointly returned at a locus and can also be a result of the large number of tissues and cell types included in the implementation of FUSION. Most hits are found in tissues without any clear relevance to the trait and absent in relevant tissues—LDL, for example, has more TWAS associations between expression and eQTL in prostate adenocarcinoma (24 genes associated), brain prefrontal cortex (23 genes associated), and transformed fibroblasts (21 genes associated) than it does in adipose (16 genes associated), blood (11 genes associated), or liver (5 genes associated). Individual genes were often identified as hits in multiple tissues, but with an inconsistent direction of effect—that is, increased gene expression correlated with an increase in the quantitative trait or disease risk in some tissues, but a decrease in others, which suggests that the gene in question may not be the one whose expression contributes to the complex trait. Because of this possibility and the known biological role of many of our genes, we restricted our results to tissues with established relevance to our traits.

## Fine-mapping GWAS hits

We fine-mapped the GWAS variants located within ±100 kb of our putatively causative genes by applying the SuSiE algorithm (*Wang et al., 2020b*) on the unconditional summary statistics from the GWAS of breast cancer, Crohn disease, ulcerative colitis, type II diabetes, height, LDL cholesterol, and HDL cholesterol. An LD reference panel from UKBB subjects of European ancestry was used for this

analysis. Fine-mapped variants were annotated using snpEff (v4.3t). Only non-coding variants were kept for further analysis.

## Functional genomic annotation of fine-mapped hits

We projected fine-mapped GWAS variants onto active regions of the genome, identified using three alternative approaches: (i) histone modification features, (ii) DHS, and (iii) ChromHMM enhancers.

First, we looked at three histone modification marks, namely, acetylation of histone H3 lysine 27 residues (H3K27ac), mono-methylation of histone H3 lysine 4 residues (H3K4me1), and tri-methylation of lysine 4 residues (H3K4me3) from the Roadmap Epigenomics Project (*Wainberg et al., 2019*) to identify functional enhancers, which are key contributors of tissue-specific gene regulation. We downloaded imputed narrowPeak sets for H3K27ac, H3K4me1, and H3K4me3 from the Roadmap Epigenomics Project (*Wainberg et al., 2019*) ftp site (available here) for 14 different tissue types (*Supplementary file 1*). For each tissue type, we extracted the narrow peaks that are within ±5 Mb of our putatively causative genes. Then following the approach described in Fulco et al. *Barbeira et al., 2018*, we extended the 150 bp narrow peaks by 175 bp on both sides to arrive at candidate features of 500 bp in length. All features mapping to blacklisted regions (https://sites.google.com/site/anshul kundaje/projects/blacklists) were removed (*Kundaje et al., 2015*). The remaining features were re-centered around the peak and overlapping features were merged to give the final set of features per histone modification track. The mean activity/strength of a feature ($A_F$) was calculated by taking the geometric mean of the corresponding peak strengths from H3K27ac, H3K4me1, and H3K4me3 marks. We then combined these activity measurements with the linear distances between the features and the transcription start sites of causative genes to compute ABD scores (a simplified version of ABC scores *Barbeira et al., 2018*) for gene–feature pairs using the following formula:

$$ABDscore_{F,G} = \frac{A_F \times 1/d_{F,G}}{\sum_{all f within \pm 5 Mb of G} A_f \times 1/d_{f,G}}$$

The ABD score can be thought of as a measure of the contribution of a feature, F to the combined regulatory on gene, G. A high ABD score may serve as a proxy for an increased specificity between a chromatin feature and the gene of interest. We projected the fine-mapped variants onto the chromatin features in different tissue types to assess whether there is an enrichment of likely causal GWAS hits in regulatory features near our putatively causative genes. Both proximity (genomic distance) and specificity (ABD scores) were considered to determine the regulatory contribution of the fine-mapped hits.

Next, we looked at the DHS that are considered to be generic markers of the regulatory DNA and can contain genetic variations associated with traits and diseases (*Meuleman et al., 2020*). We downloaded the index of human DHS along with biosample metadata from https://www.meuleman.org/research/dhsindex/. The index was in hg38 coordinates, which were converted to hg19 coordinates using the online version of the hgLiftOver package (https://genome.ucsc.edu/cgi-bin/hgLiftOver). We created a DHS index for each tissue type relevant to the traits and diseases we analyzed by including all DHS that are present in at least one biosample from a certain tissue type (*Supplementary file 2*). We then selected DHS that lie within ±100 kb of the TSS of our putatively causative genes. Since DHS are of variable widths, we recentered the summits in a 350 bp window and merged overlapping sites in the same way as we did for other chromatin marks. We calculated the mean activity ($A_F$) by averaging the strengths of all the merged sites. Next, we calculated the activity by distance score for each DHS and gene pair using the same formula described above. Finally, for each fine-mapped variant, we identified all DHS that fall within ±100 kb of the variant.

Finally, we used in silico chromatin state predictions (chromHMM core 15-state model *Wainberg et al., 2019*) for relevant tissue types (*Supplementary file 1*) to identify active enhancer regions in the genome. Tissue-specific chromHMM annotations were downloaded from the Roadmap Epigenomics Project *Wainberg et al., 2019* ftp site (available here). We considered a fine-mapped variant to fall in an enhancer region if it was housed within a chromHMM segment described as either *enhancer*, or *bivalent enhancer*, or *genic enhancer*. Since chromHMM annotations are not accompanied by activity measurements, the ABD approach could not be applied here.

## Acknowledgements

We thank Alkes Price, Alex Bloemendal, Benjamin Neale, Bogdan Pasanuic, Sasha (Alexander) Gusev, and Matt Warman for their helpful discussions. This research was supported by NIH grants R35GM127131, R01HG010372, R01MH101244, and U01HG012009. NJC was supported by NIH training grant T32GM74897. UK Biobank was accessed under projects 14048 and 10438. TOPMed data were used under dbGaP project 28674.

## Additional information

### Funding

| Funder | Grant reference number | Author |
|---|---|---|
| National Institutes of Health | R35GM127131 | Shamil R Sunyaev |
| National Institutes of Health | R01HG010372 | Shamil R Sunyaev |
| National Institutes of Health | R01MH101244 | Shamil R Sunyaev |
| National Institutes of Health | U01HG012009 | Chris Cotsapas |
| National Institutes of Health | T32GM74897 | Shamil R Sunyaev |

The funders had no role in study design, data collection and interpretation, or the decision to submit the work for publication.

### Author contributions

Noah J Connally, Conceptualization, Data curation, Formal analysis, Investigation, Visualization, Methodology, Writing – original draft, Writing – review and editing; Sumaiya Nazeen, Data curation, Formal analysis, Investigation, Visualization, Methodology, Writing – review and editing; Daniel Lee, Data curation, Formal analysis, Investigation, Writing – review and editing; Huwenbo Shi, Resources, Data curation, Software, Formal analysis, Investigation, Methodology, Writing – review and editing; John Stamatoyannopoulos, Resources, Funding acquisition, Writing – review and editing; Sung Chun, Chris Cotsapas, Conceptualization, Supervision, Methodology, Writing – review and editing; Christopher A Cassa, Conceptualization, Resources, Supervision, Methodology, Project administration, Writing – review and editing; Shamil R Sunyaev, Conceptualization, Resources, Supervision, Funding acquisition, Methodology, Writing – original draft, Project administration, Writing – review and editing

### Author ORCIDs

Noah J Connally (iD) http://orcid.org/0000-0003-3818-6739
Shamil R Sunyaev (iD) http://orcid.org/0000-0001-5715-5677

### Decision letter and Author response

Decision letter https://doi.org/10.7554/eLife.74970.sa1
Author response https://doi.org/10.7554/eLife.74970.sa2

## Additional files

### Supplementary files

• Supplementary file 1. Roadmap epigenomics aliases of tissue types used for functional genomic analysis. Tissue types from the Roadmap Epigenomics Consortium do not perfectly match those from GTEx. However, there is overlap, and as with GTEx, we analyzed trait-relevant tissues.

• Supplementary file 2. Tissue types and biosamples from the DNase I hypersensitive sites index used for functional genomic analysis.
 Meuleman et al., 2020 assess DNase I hypersensitive sites across 438 cell and tissue types; we selected the above based on their relevance to our complex traits.

- Supplementary file 3. TOPMed URLs used.
- Transparent reporting form

## Data availability

Numerical data for results is included in Source Data 1. The dataset generated (GWAS summary statistics conditioned on coding variants) can be found at https://doi.org/10.5061/dryad.612jm644q. Code for this project is available at https://github.com/NJC12/missing_link_association_function (copy archived at swh:1:rev:46d9072b7cc13f6532203d1494eec4d0f634e092). UK Biobank data was used and is available by application to the UK Biobank Data Access Committee https://www.ukbiobank.ac.uk/enable-your-research/apply-for-access. TOPMed Whole Genome Sequencing Project - Freeze 5b, Phases 1 and 2 data was used and can be accessed at https://www.nhlbiwgs.org/topmed-whole-genome-sequencing-project-freeze-5b-phases-1-and-2, full list of TOPMed URLs available in Supplementary file 3.

The following dataset was generated:

| Author(s) | Year | Dataset title | Dataset URL | Database and Identifier |
|-----------|------|---------------|-------------|-------------------------|
| Connally NJ | 2022 | GWAS results conditioned on coding variants | https://dx.doi.org/10.5061/dryad.612jm644q | Dryad Digital Repository, 10.5061/dryad.612jm644q |

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

## Appendix 1

### Evidence for the relationship between Mendelian and complex traits

More generally, this expectation is supported by several lines of evidence. Comorbidity between Mendelian and complex traits has been used to identify common variants associated with the complex traits (*Blair et al., 2013*). Early GWAS found associations near genes identified through familial studies of severe disorders (*Voight et al., 2010*; *Chan et al., 2015*), and later implicated some of the same genes in complex and Mendelian forms of cardiovascular (*Kathiresan and Srivastava, 2012*) and neuropsychiatric (*Zhu et al., 2014*) traits. More recent analyses have found that GWAS associations are enriched in regions near causative genes for cognate Mendelian traits in blood traits (*Vuckovic et al., 2020*), lipid traits and diabetes (*Weiner et al., 2022*), as well as a diverse collection of 62 traits (*Freund et al., 2018*). Another recent method used transcriptomic, proteomic, and epigenomic data to prioritize genes and found that, in a selection of nine phenotypes, the selected genes were enriched for Mendelian genes causing similar traits (*Mountjoy et al., 2021*). Together, these suggest that genes causing Mendelian traits also influence cognate complex traits, but not through the same coding mechanism.

Genes can also harbor coding variants tied to less severe forms of a trait. These coding variants are more difficult to identify individually as their effect sizes are much smaller. However, the greater number of variants (in aggregate) and freedom from searching for severe segregating traits allows the use of large population datasets. *Backman et al., 2021* used burden testing on UK Biobank data to identify genes whose coding variation affects complex traits, finding many genes not identified through familial studies.

