## [Editor Report]

The findings reported here are important because they address the issue of how complex traits arise from their genetic underpinnings. There is an assumption that genetically mediated variation in transcript abundance, usually detected via analysis of expressed quantitative trait loci, is key to this process, but we lack robust evidence in support of that view. This article finds limited evidence that the baseline expression of trait-related genes explains the associations between complex traits and genetic variants (as identified from genome-wide association studies), leading to the view that the field needs to confront a problem of 'missing regulation.'

---

## [Decision Letter]

**Decision letter after peer review:**

Thank you for submitting your article "The missing link between genetic association and regulatory function" for consideration by *eLife*. Your article has been reviewed by 3 peer reviewers, and the evaluation has been overseen by a Reviewing Editor and Molly Przeworski as the Senior Editor. The reviewers have opted to remain anonymous.

Essential revisions:

All the reviewers agreed that you've identified important concerns regarding the hypothesis that the majority of genetic variants contributing to complex traits act by the altering transcript abundance. However we would like you to address the following points:

1) Most importantly, we're not convinced you've chosen an appropriate set of genes to make your case. Your assumption that Mendelian and cognate complex traits share the same set of causal genes needs further justification. One possibility might be to use as a starting point the association between GWAS and rare variant analyses reported in the UK biobank data

https://doi.org/10.1038/s41586-021-04103-z

and/or data from the recent study of obesity (https://www.science.org/doi/10.1126/). The enrichments reported in those papers are higher than the Cell paper you quote, and would provide a set of genes that are not selected on the basis of Mendelian large effects.

2) If you choose to focus on the 143 genes from Mendelian genetics then there are a number of issues you need to resolve.

(i) why are candidate genes were not enriched in the GWAS regions for height and breast cancer?

(ii) we suggest you quantify the number of genes in the GWAS regions expected to be found if the 143 genes had been randomly selected. Correcting the observed number of genes for that expected by chance (e.g., subtracting the observed number by that expected by chance), the proportion of candidate genes in the GWAS regions is likely to be small.

3) Tissue context. The tissues listed in Table 2 omit some that are relevant to the phenotype (such as bone for height (Finucane et al. 2015 NG)). Expanding this list and selecting appropriate tissues might substantially alter your conclusions. Among the 84 putatively causative genes that overlap GWAS signals, the number that overlap is reduced substantially when restricting analysis to the selected tissues for each trait. If genes function only in the relevant tissues, using bulk expression data loses power, but is unlikely to give false positives. Thus, it is possible that for the traits analysed, not all relevant tissues are selected, so that only a fraction of genes identified in the bulk expression analysis can be replicated in the tissue-specific analysis.

4) How much do both LD differences between GWAS and eQTL samples and the presence of allelic heterogeneity contribute to the observed low colocalization rate? While we agree that power for locus detection is probably not a big issue, sample size differences betweenGWAS and GTEx datasets might make small differences in LD between the two samples cause a statistical separation of the signals, even when trait phenotype and gene expression truly share a causal variant.

The presence of more than one causal variant with allelic heterogeneity at the locus may also play a part in the failure of colocalization. Consider two causal variants for the complex trait, one regulating the target gene and the other regulating another gene in co-expression. Potentially, the presence of the second causal variant would diminish the colocalization probability at the target gene.

One possible way to deal with these issues is to perform simulations to quantify the influence of tissue-specific expression effects, LD differences between eQTL and well-powered GWAS, and allelic heterogeneity. At the least, we think you should discuss the problems we raise here in some detail.

5) TWAS results. While only 6% of the putatively causative genes are identified by TWAS with the correct direction of effect, this number is misleading as one may interpret it as meaning that only 6% of the functionally relevant genes are regulated by trait-associated variants. In fact, 46% of the genes are detected by TWAS, but only 11% are confirmed in their selected tissues, among which about half (5/9) have correct direction of effect. The result could be due to the selection of relevant tissues, or it may reflect a nonlinear relationship between expression and trait or the presence of cell type heterogeneity within a tissue. Again we think a more nuanced discussion of this issue is important.

*Reviewer #1 (Recommendations for the authors):*

The number of traits examined (seven or nine) limits the generalizability of the findings. The study would strongly benefit from adding more traits. In addition, the study would benefit from the inclusion of results from the few available cell-type-specific/dependent eQTLs from single-cell or deconvolved bulk RNA-Seq experiments.

---

## [Author Response]

Essential revisions:All the reviewers agreed that you've identified important concerns regarding the hypothesis that the majority of genetic variants contributing to complex traits act by the altering transcript abundance. However we would like you to address the following points:1) Most importantly, we're not convinced you've chosen an appropriate set of genes to make your case. Your assumption that Mendelian and cognate complex traits share the same set of causal genes needs further justification. One possibility might be to use as a starting point the association between GWAS and rare variant analyses reported in the UK biobank datahttps://doi.org/10.1038/s41586-021-04103-zand/or data from the recent study of obesity (https://www.science.org/doi/10.1126/). The enrichments reported in those papers are higher than the Cell paper you quote, and would provide a set of genes that are not selected on the basis of Mendelian large effects.

We believe that the genes we have selected have a high probability of mediating GWAS associations, given both their relevance to cognate traits and their enriched proximity to GWAS peaks. But the reviewer suggestion to incorporate genes from Backman et al. is a good one. In addition to providing additional genes to test, the genes from Backman et al. are identified based on analysis of the same complex traits—not their Mendelian cognates. These results add 90 unique genes to our analysis, and these genes, like the ones we selected, have significant enrichment near GWAS peaks. In evaluating the colocalization and TWAS results for this set of genes, we replicate and strengthen the results found using our original set of genes.

2) If you choose to focus on the 143 genes from Mendelian genetics then there are a number of issues you need to resolve.(i) why are candidate genes were not enriched in the GWAS regions for height and breast cancer?(ii) we suggest you quantify the number of genes in the GWAS regions expected to be found if the 143 genes had been randomly selected. Correcting the observed number of genes for that expected by chance (e.g., subtracting the observed number by that expected by chance), the proportion of candidate genes in the GWAS regions is likely to be small.

(i) The reviewers’ notes about enrichment—and its absence in height and BC—prompted us to review our analysis of it. The enrichment for five of our phenotypes remained significant, and the lack of enrichment for breast cancer genes proved artifactual. After accounting for the artifact, the enrichment of breast cancer genes displays the same pattern as most other phenotypes, displaying highly significant enrichment as compared to the genomic background and a permutation analysis. Figure 1 Supp. 1 has been updated to reflect this change, and to add the enrichments found in Backman et al.

Because our original analysis of height has nominal, but not corrected, significance for enrichment, the problem may be one of power. The set of height genes identified by Backman et al. is larger than our original set and displays a significant enrichment in proximity to GWAS signal. This enrichment is also present when the two gene sets are combined, as shown in the updated Figure 1 Supp. 1.

(ii) Taking random sets of genes, or the entire set of non-putatively-causative genes shows that, given the size of our gene set, we would expect 43 randomly selected genes to fall within 1 Mb of a peak (95% confidence interval: 31.5-54.5). Instead, we find 147 peak-adjacent genes. When looking closer to genes, the enrichment increases. At a distance of 100 kb, we find 104 putatively causative genes, but the null model predicts only 11 (95% CI 4.5-17.0), a roughly ten-fold difference.

Enrichment remains significant even when using a more conservative null. It may be that genes like ours, with importance to phenotype, are more likely than random genes to fall near GWAS peaks, even if their phenotype does not correspond to the GWAS phenotype. In this case, we might see enrichment even in the absence of a relationship between our Mendelian and complex traits. To account for this, we also tested significance by testing genes sets against different phenotypes (e.g. testing our LDL genes with a UC GWAS, and our height genes with a T2D GWAS). The results of this permutation are visible in Figure 1 Supp. 1, and further confirm the enrichment.

Finally, non-expression based analysis found that Mendelian genes had large enrichments in heritability. As in our study, they included Mendelian genes for diabetes and LDL—the Mendelian diabetes genes were enriched 65-fold for common-variant heritability and the Mendelian LDL genes were enriched 212-fold (Weiner et al. 2022).

3) Tissue context. The tissues listed in Table 2 omit some that are relevant to the phenotype (such as bone for height (Finucane et al. 2015 NG)). Expanding this list and selecting appropriate tissues might substantially alter your conclusions. Among the 84 putatively causative genes that overlap GWAS signals, the number that overlap is reduced substantially when restricting analysis to the selected tissues for each trait. If genes function only in the relevant tissues, using bulk expression data loses power, but is unlikely to give false positives. Thus, it is possible that for the traits analysed, not all relevant tissues are selected, so that only a fraction of genes identified in the bulk expression analysis can be replicated in the tissue-specific analysis.

In our initial submission, we had been reluctant to expand the list of tissues for two reasons. First, increasing from the small number of tissues with known biological relevance to all tissues (or all non-brain tissues) increases the multiple-testing correction burden. Second, and, in our eyes, more important, colocalizations in tissues without clear biological relevance are not biologically interprable. Such hits can be results of complicated genetic architecture (e.g. shared eQTLs), power differences in tissues with correlated expression, or biology not directly related to the trait in question.

That said, the tissue data we have access to are incomplete, and we are without question missing some relevant tissues. Additionally, some relevant tissues have lower sample sizes, and thus lower power, than tissues that are not relevant but may still share eQTLs. To overcome these problems, we applied Multivariate Adaptive Shrinkage (MASH), a Bayesian method that detects correlations between different (in this case tissues) and uses them to produce posterior estimates of summary statistics in each tissue (Urbut et al. 2019). Unlike meta-analysis, which produces one result, the effect size estimates for each tissue are distinct, though informed by one another.

Using MASH has a pronounced effect on colocalization results. The number of non-putatively causative genes colocalizing increases from 389 to 489, while the number of putatively causative genes in our Mendelian set is unchanged, remaining at 2. The number of genes from the Backman et al. set increases from 2 to 5. Though this is a proportionally large increase, it still represents a small fraction of genes. We have updated our paper to use these results—which should be less dependent on the tissues we selected—but the message has not changed.

4) How much do both LD differences between GWAS and eQTL samples and the presence of allelic heterogeneity contribute to the observed low colocalization rate? While we agree that power for locus detection is probably not a big issue, sample size differences betweenGWAS and GTEx datasets might make small differences in LD between the two samples cause a statistical separation of the signals, even when trait phenotype and gene expression truly share a causal variant.The presence of more than one causal variant with allelic heterogeneity at the locus may also play a part in the failure of colocalization. Consider two causal variants for the complex trait, one regulating the target gene and the other regulating another gene in co-expression. Potentially, the presence of the second causal variant would diminish the colocalization probability at the target gene.One possible way to deal with these issues is to perform simulations to quantify the influence of tissue-specific expression effects, LD differences between eQTL and well-powered GWAS, and allelic heterogeneity. At the least, we think you should discuss the problems we raise here in some detail.

The ability of our statistical tools to actually find colocalizations is a critical one in this project. Small sample size increases the variance of the LD matrix, but is one of only many factors that influence power, which include LD differences between study populations and eQTL effect sizes.

Though we restricted both GWAS and GTEx samples to subjects with European ancestry and used PCs as covariates, reviewers are correct that there are likely to be LD differences between samples, due to both slight variations in populations and the smaller sample sizes of GTEx. Analysis of colocalization tools in cases of mismatched LD have shown that decreases in power are small. Chun et al. (2017) tested JLIM in simulated conditions of modest population mismatch, using CEU haplotypes to create the GWAS, and haplotypes from all non-Finnish Europeans for eQTL associations. They then attempted to distinguish shared vs. distinct causative variants for GWAS and eQTL, finding no decrease in sensitivity or specificity (Supp. Figure 6 of Chun et al. 2017).

The case in which two genes are co-regulated by nearby variants, both causative for the GWAS trait, creates a condition of allelic heterogeneity for the GWAS trait (as opposed to the expression trait). Chun et al. evaluated JLIM’s loss of power as a result of AH, and found that the power loss is small, except in cases in which the two variants have equal effects (Supp. Figure 10 of Chun et al. 2017). Testing cases in which the AH occurs for the expression trait returned a similar result (Supp. Figure 9 of Chun et al. 2017).

Hukku et al. (2021) performed similar analyses on coloc, eCAVIAR, and fastENLOC. Allelic heterogeneity was found to damage the power of coloc (by about a factor of 2). Testing on different pairs of populations, they conclude that extreme LD mismatches (e.g. Finnish vs. Yoruban samples) can lead to substantial power loss, but moderate LD mismatches (e.g. Finnish vs. British samples) do not. Though a factor of two is substantial, it would not change the qualitative conclusions of this paper. Overall, given the variety of methods we employ (including those, such as JLIM, more robust to AH), we are confident that they have, when taken together, been shown to be robust to the concerns raised.

Finally, TWAS should, by design, be less vulnerable to LD differences and allelic heterogeneity. This can result in false positives, when genes with correlated expression are identified together, despite only one being causative. It can also result in non-causative genes being prioritized over causative ones, however, generally both genes will be identified (Wainberg et al. 2019).

5) TWAS results. While only 6% of the putatively causative genes are identified by TWAS with the correct direction of effect, this number is misleading as one may interpret it as meaning that only 6% of the functionally relevant genes are regulated by trait-associated variants. In fact, 46% of the genes are detected by TWAS, but only 11% are confirmed in their selected tissues, among which about half (5/9) have correct direction of effect. The result could be due to the selection of relevant tissues, or it may reflect a nonlinear relationship between expression and trait or the presence of cell type heterogeneity within a tissue. Again we think a more nuanced discussion of this issue is important.

We agree that the dropoff between the 46% of putatively causative genes identified by TWAS broadly, and the 6% identified with more stringent criteria, is quite striking. However, we also believe that a degree of conservatism is called for when employing TWAS. Not only is TWAS vulnerable to false positives caused by linkage disequilibrium between distinct eQTL and GWAS variants, but each TWAS hit is actually a set of “joint genes'' that collectively display a relationship between expression and GWAS phenotype, but cannot be distinguished between.

Consequently, if taken at face value across all tissues, TWAS results in a large number of associations. For height, 4,850 unique genes fall within the implicated joint sets. For LDL, 154 unique genes do.

For example, the *APOE* gene falls next to a GWAS peak for LDL, and TWAS finds that its expression is associated with LDL. However, it is one member of a joint set at the locus—a joint set containing 14 genes. If its association was in a tissue with known relevance to LDL cholesterol levels, we might feel comfortable ascribing this GWAS peak at least in part to the eQTL for *APOE*. However, its association is found only in lung squamous cell carcinoma tissue from The Cancer Genome Atlas. This may reflect a real functional understanding, once whose existence is missed in our tissues for any of the reasons outlined above. But, given the tissue, we are simply not confident enough in this association to use it as a benchmark for evaluating methods or drawing strong conclusions about biology.

This type of result is common in TWAS. In the example above, we note the difficulty in interpreting an association between *APOE*'s expression and LDL levels that is found in a seemingly unrelated tissue (lung squamous cell carcinoma), but not liver, adipose, or blood. As a matter of fact, TWAS finds more associations between expression and eQTL in prostate adenocarcinoma (24 genes associated), brain pre-frontal cortex (23 genes associated), and transformed fibroblasts (21 genes associated) than it does in adipose (16 genes associated), blood (11 genes associated), or liver (5 genes associated). Including all these results would leave us unsure of what we were actually detecting, as well as what its significance is.

Reviewer #1 (Recommendations for the authors):The number of traits examined (seven or nine) limits the generalizability of the findings. The study would strongly benefit from adding more traits. In addition, the study would benefit from the inclusion of results from the few available cell-type-specific/dependent eQTLs from single-cell or deconvolved bulk RNA-Seq experiments.

We selected phenotypes to reflect a broad swathe of the qualities displayed by complex traits. The research includes quantitative and binary traits. They cover immunity, metabolism, and neoplasm phenotypes, as well as the classic anthropomorphic trait, height.

Reviewers are correct that many genes have cell type specific eQTLs, and that this is likely true of our gene sets as well. These eQTLs may be missed in bulk tissue expression analysis. However, our argument is not based on genes for which we have *not found* eQTLs; it is based on genes for which we *have found* eQTLs, but whose eQTLs do not explain GWAS peaks. Evidence indicates that incorporating single-cell data might make additional eQTLs “appear,” increasing the number of genes in our analysis, but any model that does not also explain the “disappearance” of discovered eQTLs is incompatible with our observations.